# The Grammar of a Polysemantic Neuron: Understanding How Neurons Compress Multiple Concepts

## Abstract

One of the pivotal recent challenges in the field of neural network interpretability is polysemanticity, where a single neuron is activated by multiple, often unrelated concepts. This phenomenon obstructs a straightforward functional understanding of individual neurons. Despite its significance, polysemanticity has not yet been examined in a systematic and comprehensive manner. In this paper, we provide the first in-depth analysis of how polysemanticity emerges across different architectures and layers. Our key contributions are as follows: (1) we introduce effective methods to disentangle visual concept clusters encoded within a single neuron across diverse model architectures; and (2) using this approach, we conduct the systematic investigation of polysemanticity, spanning from its properties across models to the pathways underlying its formation. We believe that this work underscores the necessity of shifting the unit of analysis from individual neurons to the concept clusters they encode.

## 1 Introduction

Deep learning models have achieved remarkable performance across a wide range of applications, but their rapidly increasing complexity and scale have made it ever more difficult to understand their underlying mechanisms (Arrieta et al., 2020). As a result, interpretability research—which seeks to clarify the internal representations and computational processes learned by models—has become increasingly important and is now an active area of study. In particular, mechanistic interpretability, which examines the functions of neurons and modules in storing and processing knowledge, has emerged as one of the most prominent research directions in the field (Olah et al., 2017; 2020; Bereska & Gavves, 2024; Elhage et al., 2021).

In the vision domain, such research has been actively pursued, with approaches like Bau et al. (2017); Oikarinen & Weng (2022) analyzing models at the level of individual neurons by identifying shared visual features among the samples that strongly activate them. This approach rests on the implicit assumption that each neuron corresponds to a single function. Recent studies, however, have shown that multiple concepts often coexist within a single neuron—a phenomenon known as ***polysemanticity*** (Elhage et al., 2022; Dreyer et al., 2024). As a result, single-neuron analysis faces fundamental limitations in faithfulness, and such interpretations alone are insufficient to fully explain the underlying mechanisms of deep models.

To account for polysemanticity, several approaches have been proposed, such as training interpretable Sparse Autoencoders (SAEs) to indirectly explain internal features of foundation models, or tracing the circuit paths involved in specific decisions under the assumption of distributed representation (Cunningham et al., 2023; Thasarathan et al., 2025; Zaigrajew et al., 2025; Kwon et al., 2025; Rajaram et al., 2024; Wang et al., 2022). However, SAE-based approaches have the limitation that they do not directly reveal the intrinsic structural flow of the network, while circuit-based approaches are largely confined to sample-specific analyses and have been studied mainly in the language domain, thus remaining at an early stage for uncovering the global mechanisms of a model. There has been some work attempting to measure and analyze polysemanticity (Dreyer et al., 2024; Yu et al., 2025; Sawmya et al., 2024; Hesse et al., 2025), but these approaches either suffer from limited applicability—for example, by assumptions such as neuron separability constraints or by

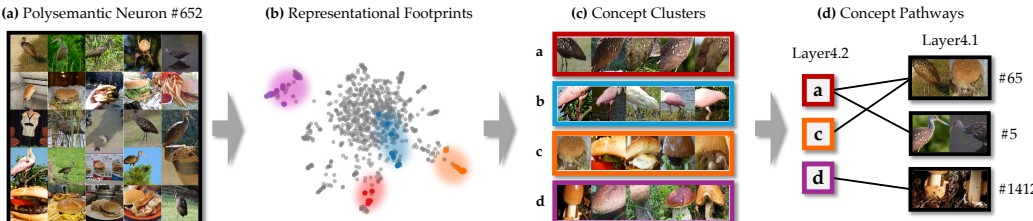

**(a)** Polysemantic Neuron #652  **(b)** Representational Footprints  **(c)** Concept Clusters  **(d)** Concept Pathways

Figure 1: An overview of our methodology for disentangling concepts within a polysemantic neuron and tracing their compositional origins. **(a)** We begin with a polysemantic neuron (e.g., ResNet-50, Layer4.2, #65) that responds to a diverse set of images. **(b)** For each activating image, we compute its representational footprint using attribution methods. These footprints, projected here by UMAP, reveal a clear cluster structure. **(c)** Clustering these footprints successfully disentangles the neuron's mixed selectivity into coherent conceptual groups (e.g., distinct types of birds and mushrooms). **(d)** By analyzing the origin of each concept cluster, our method naturally reveals the underlying concept pathways, showing how the neuron's selectivity is constructed from concepts in preceding layers.

restrictions on model architectures—or fall short of providing a deeper analysis of polysemanticity itself. Given these limitations, prior research has primarily focused on confirming the existence of polysemanticity or attempting to separate it indirectly, while its internal organization and structural mechanisms remain only partially understood.

Therefore, in this work, we aim to provide a comprehensive understanding of polysemantic neurons by disentangling the visual concept clusters of individual neurons based on their attribution footprints. The proposed method can be applied across diverse models in both efficient and effective way, enabling a systematic investigation of the formation patterns of polysemanticity and their internal structure. (See Figure 1.) Beyond extending interpretability, this analysis offers novel insights into the fundamental principles by which deep vision models organize and represent concepts.

## 2 How Can We Disentangle Learned Concepts?

To analyze the concepts learned by a neural network, this section details a two-step methodology. We begin by using attribution maps to characterize the representational footprint of individual concepts. Subsequently, we cluster these maps to disentangle concepts arising from distinct pathways, allowing for an analysis of their origins.

**Identifying Representational Footprints via Attribution**    The visual concept captured by a single neuron is often reflected in the common features of its most highly activated samples across a given dataset (Bau et al., 2017; Kalibhat et al., 2023; Oikarinen & Weng, 2023). While this approach identifies what a neuron has learned, in the presence of polysemanticity it is challenging to disentangle the multiple concepts activated within the same neuron. To further understand how these concepts are constructed from preceding layers, it is necessary to trace their compositional origins. To this end, we define a concept's representational footprint using attribution methods, a technique widely employed in CNN analysis to map the hierarchical flow of information (Shrikumar et al., 2017; Selvaraju et al., 2017; Simonyan et al., 2014). These methods explain a neuron's activation by propagating its signal backwards, thereby quantifying the contribution of lower-level features. This principle has been successfully applied in various attribution methods, from LRP-based techniques like CRP (Achtibat et al., 2023; Dreyer et al., 2024) to gradient-based approaches (Hesse et al., 2025).

Formally, given a dataset $\mathcal{D} = \{x_i\}$ and a network $f$, we first interpret the concept of a neuron $u^l$ in layer $l$ through its set of most highly activating samples, $\mathcal{D}_{u^l} \subset \mathcal{D}$. We define a *neuron* based on the architecture: for CNNs, we consider each channel as a neuron, as it acts as a distinct feature detector

across spatial locations [1]. For Transformers, we consider each hidden representation as a neuron, as it maintains a distinct representation for each input patch.

Then, to trace the origin of the visual concept in $u^l$, we quantify how each neuron $u_j^{l-1}$ in the preceding layer, contributes to the activation of neuron $u^l$, denoted $a_{u^l}$. The attribution $A(u_j^{l-1} \to u^l)$ was obtained following the Input×Gradient method (Shrikumar et al., 2017). For a given input $x$, the attribution is calculated as:

$$A(u_j^{l-1} \to u^l) = a_{u_j^{l-1}}(x) \odot \frac{\partial a_{u^l}(x)}{\partial a_{u_j^{l-1}}(x)} \tag{1}$$

where $\odot$ denotes the element-wise product. Since the resulting attribution from each feature $u_j^{l-1}$ is a spatial map of dimensions–$H \times W$ for CNN-based networks or the number of patches $P$ for Transformer-based networks, we aggregate these values into a single scalar representing the total contribution by summing over all spatial locations. To mitigate anisotropy, which refers to the concentration of representations along a small number of dominant directions and has been widely reported in transformer-based models, and to emphasize the distinctiveness of each attribution score, we normalize the attribution values by subtracting their mean and dividing by their standard deviation (Elhage et al., 2023; Godey et al., 2024). For convenience, we will abbreviate $A(u_j^{l-1} \to u^l)$ into $A_j^{l-1}$.

**Disentangling Concepts by Clustering Footprints**  Having defined the representational footprints, we now introduce our method for disentangling them into distinct concepts. Rather than assuming a fixed number of underlying concepts, we propose an iterative algorithm that progressively identifies and separates cohesive groups of footprints. As detailed in Algorithm 1, the process begins by attempting to partition the current set of attributions $\mathcal{A}_u = \{A_j^{l-1} | j = [1, ..., d]\}$, where $d$ denotes the number of neurons in layer $l - 1$, into $k = 2$ clusters. A cluster $\mathcal{C}$ is considered a cohesive concept if its internal coherence, which we define as the average pairwise cosine similarity between its members, exceeds an adaptive threshold $\tau$. Cohesive concepts are accepted as final and removed from the working set $\mathcal{A}'$. The algorithm then dynamically adjusts the number of clusters $k$ and repeats this process on the remaining attributions until no further cohesive concepts can be disentangled.

A key component is the dynamic calculation of this cohesion threshold $\tau$. A fixed, universal threshold would fail to adapt to the diverse similarity distributions exhibited by the footprints of different neurons. We therefore compute $\tau$ adaptively from the set of footprints currently under analysis, $\mathcal{A}'$. Our empirical analysis reveals that the character of the similarity distribution systematically changes with network depth, as illustrated in Figure 5(right). Early layers typically exhibit a unimodal, long-tailed distribution of similarities. In contrast, later layers often display a bimodal distribution, where the emergence of a second peak suggests the formation of distinct, highly cohesive conceptual groups.

To create a robust criterion that accommodates both scenarios, we define the threshold $\tau$ as the maximum of two candidates: (1) the 95th percentile of the pairwise similarity values, and (2) the value corresponding to the second peak of a Kernel Density Estimate (KDE) fitted to the similarity distribution. The percentile provides a robust baseline, particularly for the unimodal distributions in early layers, while the second peak offers a more semantically meaningful partition for the well-formed concepts in later layers. This dual-criterion approach ensures that our definition of a concept is contextually grounded across the network's hierarchy. The sensitivity analysis on the threshold parameter $\tau$ is in Appendix B.

**Comparison with Existing Work**  To demonstrate the effectiveness of our method in disentangling and explaining multiple concepts encoded within a single neuron, we compared it against the commonly used approach, PURE (Dreyer et al., 2024). Unlike ours, which flexibly determines the number of clusters per neuron and dataset, PURE requires a pre-defined number of clusters to be fixed in advance. Accordingly, in Figure 2-(a), we set the parameter k=4 for PURE and compared

---

[1]The term 'neuron' is often used interchangeably with 'feature' in the literature, or 'channel' in the context of CNNs.

---

**Algorithm 1** A Method for Iterative Concept Disentanglement

---

**Input:** A set of attribution footprints $\mathcal{A}_u$, a cohesion threshold $\tau$.
**Output:** A set of disjoint concepts $\mathcal{C}^*$, where each concept is a set of footprints.

1: **function** DISENTANGLECONCEPTS($\mathcal{A}_u, \tau$)
2:      $\mathcal{A}' \leftarrow \mathcal{A}_u$          ▷ Initialize the working set of footprints
3:      $\mathcal{C}^* \leftarrow \emptyset$          ▷ Initialize the set of discovered concepts
4:      $k \leftarrow 2$
5:      **while** $|\mathcal{A}'| \geq k$ **do**
6:          Let $\mathcal{C}_{cohesive}$ be the set of all clusters $C \in \text{Cluster}(\mathcal{A}', k)$ where $\text{Cohesion}(C) > \tau$.
7:          **if** $\mathcal{C}_{cohesive} \neq \emptyset$ **then**
8:              $\mathcal{C}^* \leftarrow \mathcal{C}^* \cup \mathcal{C}_{cohesive}$      ▷ Add cohesive clusters to the final set
9:              $\mathcal{A}' \leftarrow \mathcal{A}' \setminus \bigcup_{C \in \mathcal{C}_{cohesive}} C$     ▷ Remove their members from the working set
10:              $k \leftarrow \max(2, k - (|\mathcal{C}_{cohesive}| - 1))$
11:          **else**
12:              $k \leftarrow k + 1$      ▷ Increase search granularity if no cohesive cluster is found
13:          **end if**
14:      **end while**
15:      $\mathcal{C}^* \leftarrow \mathcal{C}^* \cup \{\{a\} \mid a \in \mathcal{A}'\}$      ▷ Treat remaining footprints as individual concepts
16:      **return** $\mathcal{C}^*$
17: **end function**

where $\text{Cohesion}(C) = \text{mean}_{a_i, a_j \in C}(\text{sim}(a_i, a_j))$

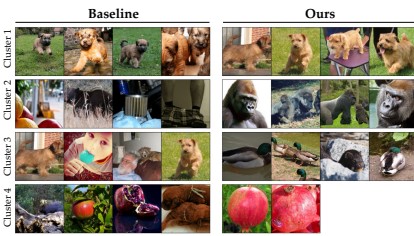

(a) 4 Clusters in ResNet50 Layer4.2 #1910.

|  | Inter-Cluster | Intra-Cluster |
|---|---|---|
| PURE | 0.79 | 0.81 |
| **Ours** | **0.41** (▼0.38) | **0.88** (▲0.07) |

(b) Average CLIP Similarity among concept clusters on ResNet50 Layer4.2.

Figure 2: Quantitative and Qualitative Comparison with Baseline (Dreyer et al. (2024)).

the extracted visual features with those obtained from our method on neuron #1910, where our approach also identified four concept clusters. Each row represents representative images of a cluster. While some clusters (e.g., cluster 1) were consistently identified by both methods, our approach uniquely revealed additional concept clusters, such as gorilla (cluster 2) and mallard duck (cluster 3). For cluster 4, our method produced a clearer and more compact visual cluster than PURE. We further performed a quantitative evaluation by comparing inter-cluster and intra-cluster CLIP similarities to assess whether the discovered clusters were semantically coherent and well separated in Figure 2-(b). Our method consistently achieved lower inter-cluster similarity and higher intra-cluster similarity, indicating superior performance in both semantic consistency and separability.

## 3 RESEARCH QUESTIONS & FINDINGS

Recent research has established polysemanticity as a fundamental property of neural networks. Prior work has largely focused on three main directions: (1) reporting its existence through empirical evidence and qualitative examples (Olah et al., 2017; 2020; Mu & Andreas, 2020); (2) investigating its theoretical underpinnings, attributing it to phenomena such as superposition (Elhage et al., 2022) or incidental causes (Lecomte et al., 2024); and (3) developing tools to mitigate or analyze it using metrics like Wasserstein distances (Sawmya et al., 2024) or clustering-based disentanglement (Dreyer et al., 2024; Hesse et al., 2025; Yu et al., 2025). Despite this extensive progress, a detailed analysis of the internal structure of polysemanticity—how different concepts are organized and relate to each

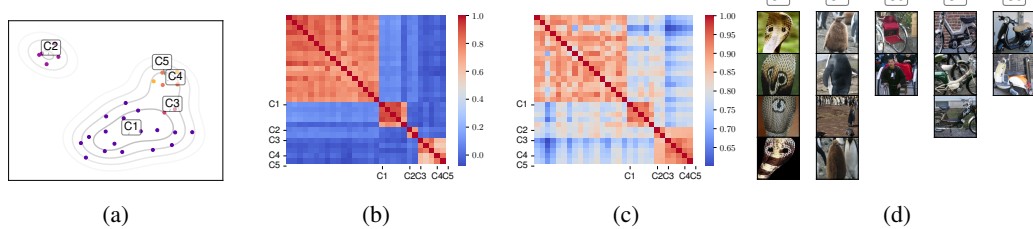

(a)        (b)        (c)        (d)

Figure 3: Dissecting a polysemantic neuron (#652) from the last transformer block of ViT-B. **(a)** UMAP embedding reveals distinct conceptual clusters within the neuron's representation footprints. **(b)** A similarity heatmap quantifies the low inter-cluster and high intra-cluster similarity. **(c)** A similarity heatmap using CLIP embeddings quantitatively shows that concepts are semantically disparate. **(d)** Exemplar images confirm the diverse and disparate visual concepts represented by each cluster. The neuron responds to diverse visual concepts such as snakeskin (C1), penguins (C2), and various two-wheeled vehicles (C3-C5).

### 3.1 Do Semantically Unrelated Concepts Cohabit in a Single Neuron?

A fundamental question in interpretability is whether a single neuron is monosemantic, representing a single concept, or polysemantic, multiplexing multiple, often unrelated, concepts. We begin with a detailed case study of a single neuron (#652) from the last block of ViT-B, as illustrated in Figure 3. Our method disentangles this neuron's activations into five distinct conceptual clusters (Figure 3a). The exemplar images in Figure 3d reveal a surprising diversity: the neuron responds to visually and semantically disparate categories, including snakeskin (C1),

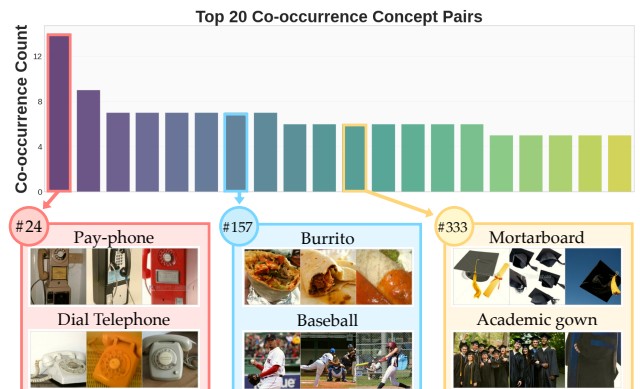

Figure 4: Top 20 co-occurrence concept pairs in ViT-B.

penguins (C2), and two-wheeled vehicles (C3-C5). The heatmap of pairwise attribution similarity (Figure 3b) confirms that these clusters are well-separated in the model's own representation space, validating our disentanglement approach. To quantitatively verify their semantic dissimilarity, we measured the similarity of the clusters in CLIP's embedding space. The resulting heatmap (Figure 3c) shows relatively low similarity between unrelated groups (e.g., C1 vs. C3), providing strong evidence that this single neuron genuinely encodes semantically distinct concepts.

Through the above analysis, we confirmed that a single neuron can contain both clusters with high embedding similarity and clusters with low similarity. Building on this, we further investigated whether certain concept pairs consistently co-occur within the model. The detailed analysis of cluster similarity and its upstream attribution paths is deferred to Section 3.3).

Focusing on the last transformer block of the ViT-B model, we identified the representative concept labels for each cluster within a neuron and tracked the frequency of co-occurrence across all concept label pairs. Figure 4 (Top) illustrates the 20 most frequent concept pairs. Three representative cases with the same co-occurrence frequency are also shown in Figure 4 (Bottom), showing diverse properties of polysemantic neurons. The pink example (neuron #24) represents phone-related concepts, even though they have distinct concept label, their semantic meanings are closely related. Similarly, the yellow example (neuron #333) The blue example (neuron #157) corresponds to the pair burrito and baseball, which are semantically unrelated. In contrast, the pink example (neuron #24)

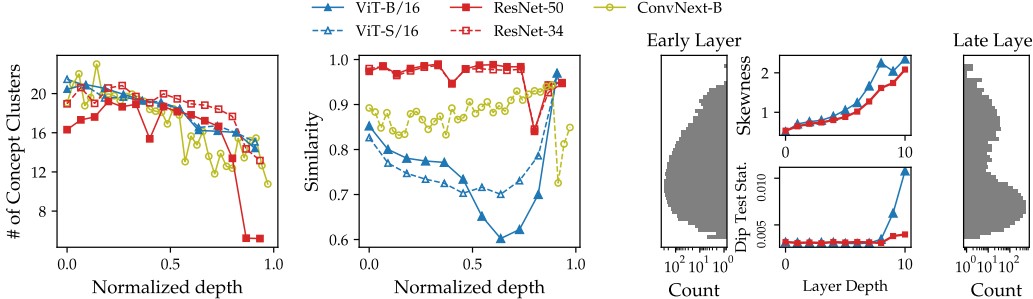

Figure 5: Polysemanticity differs across network architectures. **(Left)** The number of concepts per neuron gradually decreases in ViTs, while collapsing late in CNNs. **(Center)** Intra-cluster similarity remains high in CNNs but follows a U-shaped curve in ViTs. **(Right)** This reflects a structural evolution where the similarity distribution shifts from unimodal (early layers) to bimodal (late layers), a transition quantified by skewness and the Dip Test.

corresponds to the pair Pay-phone and Dial Telephone, which are semantically and visually highly related.

## 3.2 How Does Polysemanticity Differ across Different Networks?

Having established the existence of polysemanticity, we now investigate how its characteristics vary with network architecture and depth. Specifically, we examine two key properties: (1) the total number of concepts per neuron and (2) the internal cohesion of those concept groups.

Figure 5 (left) plots the number of disentangled concepts against normalized network depth, where we observe two distinct trends. In Vision Transformers (ViT-B, ViT-S), the number of concepts tends to gradually decrease with depth. In contrast, CNN-based architectures (ResNet, ConvNeXt) tend to maintain a high number of concepts throughout most of their layers, followed by a sharp collapse in the final blocks. This suggests that the two architectural paradigms employ different methods for information abstraction.

To understand the nature of these concepts beyond their sheer number, we measured their internal cohesion by calculating the average intra-cluster similarity (Figure 5, center). CNNs exhibit consistently high conceptual cohesion across all layers, implying that tightly grouped features are formed early on. Notably, for the ResNet models, this high cohesion is punctuated by sharp, sudden drops. These drops precisely coincide with the boundaries between major convolutional blocks (e.g., from conv2_x to conv3_x), where the spatial resolution of the feature maps is downsampled. We hypothesize that this abrupt change in resolution induces a significant shift in the representational space, leading to these transient decreases in similarity. Transformers, however, display a characteristic U-shaped curve, with lower similarity (i.e., higher diversity) in the middle layers. This suggests that ViTs may explore a wider range of feature representations before converging on highly cohesive concepts in the final stages. Figure 7 shows detailed polysemantic neurons of the lowest and highest similarity layers in each networks, which are represented in Figure 5 (center).

This difference in behavior can be explained by the evolution of the feature similarity distribution, as shown in Figure 5 (right). In early layers, all models exhibit a unimodal, skewed distribution of similarities, indicating a lack of clear conceptual separation. In later layers, this distribution becomes strongly bimodal, reflecting a clear partition into highly similar intra-concept groups and dissimilar inter-concept groups. We quantify this transition using skewness and Hartigan's Dip Test (Hartigan & Hartigan, 1985) for multimodality. The results show a clear trend of decreasing skewness and increasing multimodality with depth, a signal that is particularly strong in Transformers. The full, layer-by-layer evolution of these distributions, quantitatively supported by their Dip Test p-values, is visualized in Appendix C.

In summary, our findings reveal two distinct strategies for hierarchical feature learning. CNNs tend to maintain a large set of consistently cohesive concepts that collapse abruptly in the final layers. In

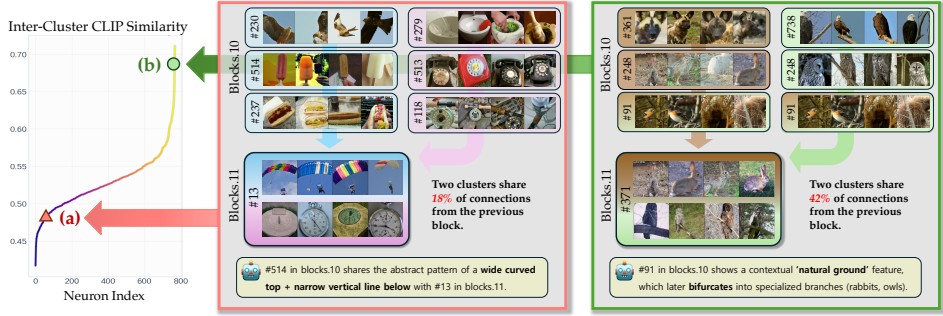

Figure 6: (**Left**) Inter-cluster similarity plot computed with CLIP embeddings, where the x-axis denotes neurons in ViT-B block 11 sorted by similarity. (**Middle**) Example of a polysemantic neuron with low inter-cluster similarity, where the trajectories diverge into separate routes. (**Right**) Polysemantic neuron with high inter-cluster similarity, where highly attributed nodes largely overlap.

contrast, Transformers undergo a process of representational diversification in their middle layers, followed by a reconsolidation into a more clearly separated set of concepts. These results suggest that the foundational inductive biases of these architectures—local, spatial convolutions versus global, token-wise self-attention—give rise to fundamentally different mechanisms for concept formation and abstraction.

### 3.3 IS ALL POLYSEMANTICITY THE SAME?

Polysemantic neurons can arise in distinct ways: some represent semantically related concepts that frequently co-occur or share local visual structures, whereas others combine entirely unrelated concepts without apparent commonality 4. A natural question is whether these two types of polysemanticity emerge through different formation mechanisms in the model. To address this, we analyze the internal pathways that give rise to polysemantic neurons.

In Figure 6-(left), we focus on the last block in ViT-B model and compute inter-cluster similarity scores using CLIP embeddings, which reveal two contrasting cases: neuron #371, whose clusters encode semantically related features, and neuron #13, whose clusters correspond to entirely unrelated concepts. To trace how these representations emerge, we visualize the three strongest attribution paths from the previous block. For instance, the parachute cluster of neuron #13 primarily derives from three upstream neurons (blue), while the clock cluster is formed from three different upstream neurons (pink). The semantics of each neuron can be conveyed by its strongly activating images: upstream neuron #230 corresponds to a blue sky, neuron #514 to a curved popsicle, and neuron #237 to an elongated hot dog. These diverse features converge into neuron #13, which activates on parachute images through shared components such as the blue background and the curved columnar shape of the parachute canopy.

Consistent interpretations are also achievable using a VLM, analogous to prior interpretation approaches Yu et al. (2025), which provide reasonable explanations when given the activating images. By supplying GPT-5 Achiam et al. (2023) with four representative images per neuron and querying the conveyed visual semantics, we obtained reasonable explanations (highlighted with the yellow box in Figure 6-(bottom)) that were consistent with human interpretation.

By contrast, the upstream contributors of the clock cluster primarily represent round-shaped objects, a semantic dimension distinct from the parachute cluster. When comparing the top-50 attributing neurons for the two clusters, only 18% overlapped, confirming that the two concepts are largely constructed from disjoint sets of pathways.

Figure 6-(right) illustrates a different type of polysemantic neuron (#371), where the two clusters exhibit semantic commonality. In this case, some upstream neurons (e.g., #91, #248) contribute to both clusters, while others (e.g., #361, #738) contribute selectively. Notably, the attribution paths overlap by as much as 42%, indicating a high degree of shared formation between clusters.

These observations suggest that polysemantic neurons are not homogeneous: some emerge through overlapping semantic circuits, while others are constructed from entirely separate pathways. This distinction highlights that the functional roles and interpretability of polysemantic neurons may vary depending on their mode of formation.

## 4    RELATED WORK

**Concept-based Interpretability**   A significant branch of interpretability research focuses on identifying human-understandable concepts within neural networks (Bau et al., 2017; Mu & Andreas, 2020; Olah et al., 2017; 2020). The goal of these methods is to associate individual neurons or directions in activation space with specific visual concepts. This research has expanded from foundational techniques to more diverse approaches, including automatically identifying concepts without human supervision (Kalibhat et al., 2023; Oikarinen & Weng, 2022; Yu et al., 2025) and extending the analysis to the level of circuits (Cao et al., 2025). A key limitation of many of these methods, however, is their implicit monosemantic assumption—that a single neuron corresponds to a single coherent concept. In practice, as we will show, many neurons encode multiple, unrelated features, an issue that undermines the reliability of their explanations.

**Theoretical understanding**   Contemporary theory reframes polysemanticity not as an error, but as an efficient coding strategy that emerges from data statistics and capacity constraints (Olah et al., 2018). For instance, work on superposition by Elhage et al. (2022) demonstrates that when a network must represent more sparse features than available dimensions, it compresses them into an overlapping code. This superposition tolerates interference because the features rarely co-occur, explaining how sparsity can lead to unrelated concepts being bound to the same neuron. Complementing this, Marshall & Kirchner (2024) approach the problem through the lens of coding theory, showing how data statistics directly influence a model's interpretability. These theoretical perspectives establish that polysemanticity is a systematic consequence of optimization, motivating the need for analysis frameworks that move beyond monosemantic assumptions.

**Polysemanticity and Disentanglement**   Building on this understanding, recent work has developed tools to directly analyze and disentangle polysemantic neurons. Prominent approaches leverage attribution methods to trace and separate concept pathways within a neuron's receptive field (Dreyer et al., 2024; Hesse et al., 2025; Yu et al., 2025). While these methods are powerful for revealing the components of a neuron's selectivity, their focus has primarily been on tool development. A persistent challenge is the difficulty in determining the true number of distinct, meaningful concepts, and a large-scale, comparative analysis of the internal structures these tools uncover has been lacking. To fill this gap, we propose a principled and data-driven framework that enables a systematic investigation and comparison of the internal structure of polysemanticity across diverse neural network architectures.

## 5    CONCLUSION

In this paper, we provide a comparative analysis with dynamic cohesion threshhold to effectively capture the dicerse similarity distribution of diferent neurons. Our systematic analysis revealed fundamentally different hierarchical learning strategies between architectures. We found that CNNs maintain a large set of consistently cohesive concepts that collapse late in the network, whereas Vision Transformers exhibit a U-shaped trend in conceptual diversity, suggesting a distinct "exploration and reconsolidation" phase. We demonstrated that this behavior is driven by the structural evolution of feature similarity distributions from unimodal to bimodal in Vision Transformers. These findings offer a new lens through which to understand the internal mechanisms of deep learning models. They suggest that architectural inductive biases lead to different strategies for concept formation and abstraction. While our analysis spans several common architectures, future work could extend this framework to other domains, such as language models, or a wider array of model scales. A particularly exciting direction is to investigate the causal link between these observed structural properties and downstream model behaviors, such as adversarial robustness or out-of-distribution generalization.

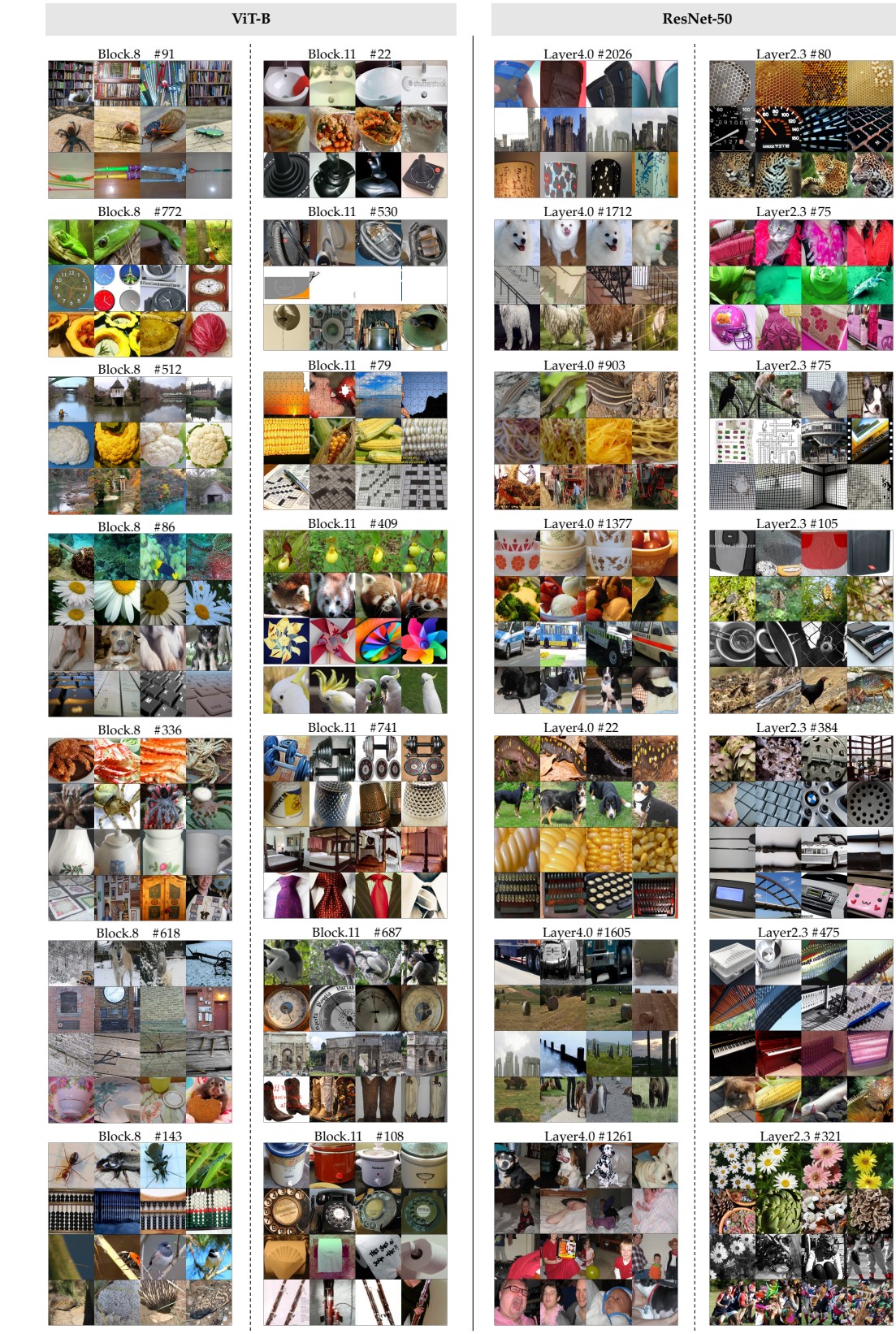

Figure 7: Examples of polysematnic neurons of the last block of ViT-B and ResNet-50. Each rows in a neuron represents a distinct concept.

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
