## A EXPERIMENTAL SETTINGS

**Models** We apply our analysis framework to a suite of widely-used models (ViT (Dosovitskiy et al., 2021), ResNet (He et al., 2015), and ConvNeXT (Liu et al., 2022)) pretrained on ImageNet (Deng et al., 2009), quantifying the diversity and semantic relationships of the concepts discovered within their neurons. Our analysis was conducted on specific layers from each model architecture. For the CNN-based models, ResNet-50 and ResNet-34, we examined layers from all bottlenecks of the main residual blocks. For the Vision Transformer models, ViT-B and ViT-S, our analysis included all 12 transformer blocks within the encoder. For ConvNext, we examined all depths(blocks) of all 4 stages.

## B THRESHOLD SENSITIVITY

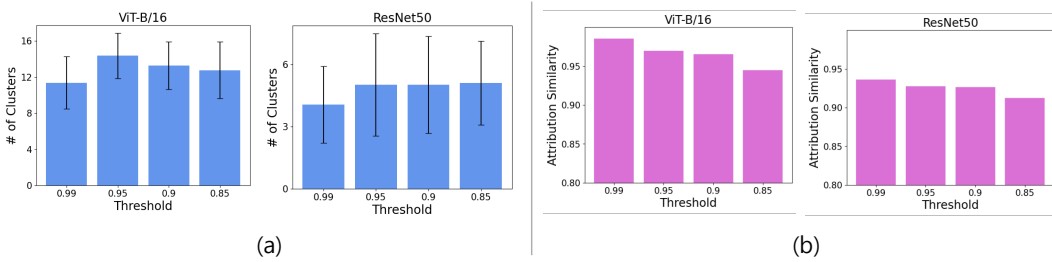

(a)                                    (b)

Figure 8: Sensitivity Analysis on Threshold.

In our proposed method, we construct flexible concept clusters tailored to each model and dataset by selecting an appropriate threshold. Specifically, we adopt the larger value between the second peak of the similarity distribution (estimated using KDE) and the score at the top-p% quantile as the threshold. To examine the sensitivity of this hyperparameter, we varied the value of p and measured its impact on both the number of clusters formed and the within-cluster similarity. The results are shown in Figure 8. We conducted experiments with p = [0.99, 0.95, 0.9, 0.85] on ResNet50 and ViT-Base-16 models. As shown in Figure 8-(a), ViT consistently produced more concept clusters on average compared to ResNet50, which aligns with prior findings that polysemantic neurons are more prevalent and complex in transformer architectures. Interestingly, as the threshold became looser, the number of clusters decreased, whereas exhibited the fewest clusters at p=0.99. This behavior arises because, at such a strict threshold, some clusters fail to form due to the distances between samples in the representation space being larger than the cutoff. Figure 8-(b) reports the within-cluster similarity scores across thresholds, all of which remain significantly high (¿0.9). A slight downward trend is observed as the threshold decreases, indicating that looser thresholds admit some noisier samples. Taken together, these results show that p=0.95 provides the best trade-off, reliably producing meaningful clusters across architectures while maintaining the highest within-cluster similarity.

## C STRUCTURAL EVOLUTION OF CONCEPTS ACROSS NETWORK DEPTH

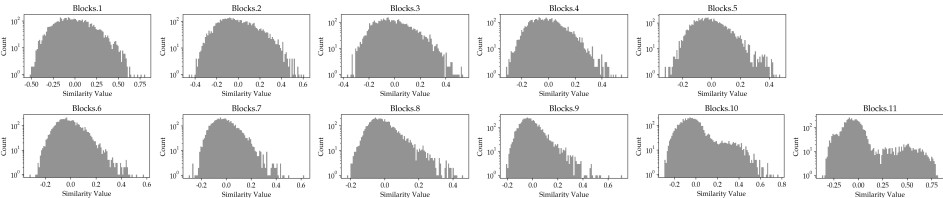

Figure 9: Evolution of the feature similarity distribution across network depth in ViT-B

The figures and analysis herein provide a more fine-grained view of the results presented in Section 3.2.

As seen in Figure 9, in early layers (a), the distribution is unimodal and skewed. As depth increases, it becomes strongly bimodal (c), indicating the separation of features into distinct, cohesive concepts.