# OpenReview forum: "The Grammar of a Polysemantic Neuron: Understanding How Neurons Compress Multiple Concepts"
_ICLR.cc/2026/Conference — ICLR 2026 Conference Withdrawn Submission_

### Official Review · Reviewer_D8wZ · 2025-10-28

**Soundness:** 2
**Presentation:** 3
**Contribution:** 2
**Rating:** 2
**Confidence:** 4

**Summary:**

The paper addresses the problem of polysemanticity in neurons, where a single neuron responds to multiple unrelated concepts. Its goal is two-fold:

1. It introduces a new method to identify concept clusters on which a polysemantic neuron activates. This is achieved by computing the attributions from the previous layer to a neuron in the layer of interest (representational footprints) and then clustering them. To avoid reliance on a predefined number of clusters, the paper proposes a dynamic clustering method based on internal coherence, defined as the average pairwise cosine similarity between cluster members.

2. It presents a study analyzing the internal structure of polysemanticity. Specifically, it investigates whether semantically unrelated concepts are encoded in the same neuron, how polysemanticity varies across models and layers, and whether different levels of polysemanticity exist.

**Strengths:**

- The paper is well written and easy to follow.
- I appreciate the motivation to analyze the internal structure of polysemanticity.
- The analysis of how polysemanticity differs across different networks and layers (Sec. 3.2) is interesting.

**Weaknesses:**

Overall, the paper is a nice read, and there are no apparent flaws in the proposed method or analysis. However, although I generally do not like using the argument of "scope" in reviews, I believe that the scope of this paper is not substantial enough to meet the acceptance criteria of ICLR. I elaborate on this reasoning below.

The paper is very ambitious in proposing both a new disentanglement method and a more in-depth analysis of the internal structure of polysemanticity. However, as a result, I feel that both components fall somewhat short in their depth and completeness.

- **W1 The proposed method is rather incremental.** As acknowledged in L. 103, the idea of representational footprints via attribution is not novel and has been explored in similar ways before. Likewise, the idea of clustering these representational footprints has precedents, for example in PURE. The main contribution of this paper lies in how the clustering is performed—specifically, in avoiding the need to select a fixed value of $k$ by instead introducing a heuristic (L. 147–154) that determines when to split a cluster. It remains unclear, however, how much the method’s performance and conclusions depend on this heuristic.

- **W2 The evaluation of the proposed method is rather sparse.** While incremental methodological contributions can be acceptable, they should be supported by a thorough evaluation and comparisons against baseline approaches to justify their significance. In this paper, however, the evaluation is limited; only Fig. 2(b) compares the proposed approach to PURE, and that only for a single setup with what appears to be a fixed value of $k$. This limited scope makes it difficult to assess the advantages and robustness of the proposed method.

- **W3 The analysis is in parts shallow.** The main conclusion in Sec. 3.1—that a single neuron can contain both clusters with high embedding similarity and clusters with low similarity (L. 259–260)—is neither particularly surprising nor novel. For instance, a similar observation can already be made from Figure 3 in PURE.

  Likewise, the main finding in Sec. 3.3—that polysemantic neurons are not homogeneous and that some emerge through overlapping semantic circuits while others arise from entirely separate pathways—is also not unexpected. Comparable insights could, for example, be drawn from Figure 4 in Hesse et al. (2025). Further, this experiment is only of a qualitative nature, and it remains debatable whether polysemanticity for related concepts is really polysemanticity or simply a more complex concept.

- **W4 The connection between the proposed method and the analysis is missing.** I would understand the dual contribution of a method and the analysis if the analysis relied on the proposed method. However, I think with PURE and a fixed $k$ similar conclusions could be made.

All in all, for the above reasons, it is difficult to take away substantial insights from the paper. It remains unclear under which circumstances the proposed method outperforms related work, and much of the analysis does not provide genuinely new findings beyond what has already been shown in prior studies.

**Minor weaknesses**

- L. 352: a "4" appears at the end of the sentence
- L. 421: dicerse --> diverse

**Questions:**

- How does the proposed method perform compared to PURE and other baseline methods across a broader range of benchmarks, layers, models, and datasets? Are there scenarios where the non-reliance on $k$ provides a clear advantage? How sensitive is the proposed method to the chosen heuristic?
- Would it be possible to conduct a similar analysis using PURE or other existing clustering approaches?
- In what ways does the analysis presented in Sec. 3.1 and Sec. 3.3 provide new insights or contributions beyond what has already been established in prior work?

I thank the authors for their effort and look forward to reading the rebuttal.

---

### Official Review · Reviewer_9FfJ · 2025-10-31

**Soundness:** 2
**Presentation:** 3
**Contribution:** 2
**Rating:** 2
**Confidence:** 4

**Summary:**

The paper studies polysemantic neurons and proposes a method to disentagle these into separate conceptual monosemantic groups. The method builds "representational footprints" using gradient times input attributions and clusters them to identify concept groups per neuron. Applying this to CNNs and Vision Transformers, the authors report structural patterns such as depth-dependent concept diversity, bimodal similarity distributions, and different modes of polysemanticity (shared vs. disjoint upstream pathways). The work provides many clear visual examples and a broad, cross-architecture survey of polysemantic behavior.

**Strengths:**

- **Important Topic:** The paper addresses an important question in interpretability: how neural networks compress multiple, semantically diverse concepts into single neurons or features, and how these can be disentangled.
- **Good Figures and Writing**: Writing and figure design are strong; qualitative examples are well chosen and effectively illustrate the intended phenomena.
- **Goog Cross-Architecture Analysis**: The cross-architecture and cross-layer comparisons (especially Section 3.2) extend previous analyses and highlight interesting hypotheses.

**Weaknesses:**

- **Limited methodological novelty**: The presented method closely mirrors the (cited) PURE method, which also clusters neuron-level attributions using gradient times input to produce monosemantic circuits and appears to lend the conceptual framing for the paper. The algorithmic change seems to be an exchange of a fixed k-means clustering algorithm in PURE for an dynamic threshold-based clustering. While a removal of limitations on prior work is important, a method that is largely adopted from prior work should adequately reference this conceptual similarity and the work should position itself clearly as a follow-up or extension of PURE. This seems especially relevant, as the introduction of the method is listed in the abstract as one of the two main contributions of this paper.

- **Unclear or limited Novelty of Findings**: The empirical results, especially Section 3.1, largely replicate insights already shown in PURE and related work. The paper presents the identification and disentanglement of polysemantic neurons as a new finding, although this phenomenon has been both theoretically explained (e.g., Elhage et al.) and empirically demonstrated in PURE with similar qualitative examples and CLIP-based analyses.

- **Lack of Quantitative Experiments**: Section 3.1 relies mostly on anecdotal evidence of handpicked neurons but shows limited quantitative evidence. While this can highlight the existence of patterns, and the examples are well chosen, the work lacks thorough quantitative experiments. Exempt from this is section 3.2, which (as stated above) presents some good quantitative cross-model and cross-layer comparisons. The experiments of Section 3.3 again mostly rely on anecdotal evidence and are only performed for a single ViT Block. Expanding the quantitative analysis would improve the papers contribution.

**Questions:**

- How does the Inter-Cluster CLIP similarity of the concepts in the concept co-occurence look across architecture and layers?
- How does the computational cost of the proposed dynamic clustering compare to PURE’s fixed k-means?
- Have alternative attribution methods (e.g. Integrated Gradients, LRP, etc.) been tested, and do results remain consistent?
- Does the improvement in average CLIP similarity between PURE and the proposed method (Fig. 2b) hold across other layers and models?

Comments:
- L352: stray “4” at sentence end.
- L419–L421: contains spelling errors.
- The Related Work section appears relocated to the end and still contains forward-referencing phrases (e.g., “as we will show”).

---

### Official Review · Reviewer_PRUD · 2025-10-31

**Soundness:** 3
**Presentation:** 2
**Contribution:** 2
**Rating:** 6
**Confidence:** 3

**Summary:**

This paper examines different forms of vision network (both convolutional and vision transformer) to understand the dynamics of polysemanticity within networks. In addition to trying to understand global dynamics of polysemanticity, they also look at the qualitative characteristics of polysemanticity at different network depths, and between convolutional networks vs vision transformers. Their method relies on first calculating "attribution footprints" (which capture how much a neuron at layer i-1 impacts a neuron at layer i, by multiplying together activation magnitude and gradient), and then clustering the attribution footprints within a neuron (across all the prior-layer neurons). Each cluster was then understood to represent a coherent concept within the network.

**Strengths:**

- I appreciated the qualitative deep dive into cluster coherence, and the discussion of the difference between semantically similar and semantically distinct clusters
- The approach to dynamically determine a number of clusters per neuron seems novel and valuable

**Weaknesses:**

- I would have preferred the paper say in the abstract that it was only focused on vision use cases
- It feels like many of the points made by this paper are ones I have heard before, and the idea of looking at attribution patterns to understand neuron concepts doesn't feel inherently novel. I would prefer that there is clearer upfront articulation of what aspects of the paper's results are new
- I found it confusing to understand and internalize what the paper was doing, and what it meant to cluster attributions (initially I thought the clustering was occurring across neurons, and it was only after coming back and looking multiple times that I de-confused myself)

**Questions:**

- Do you believe your method could work on any network types outside of vision?
- I don't understand what point is being made in section 3.3, since it devolves into very detailed explanations of individual neurons rather than leading with a broad intuitive framing.

---

### Official Review · Reviewer_2pvM · 2025-11-01

**Soundness:** 3
**Presentation:** 3
**Contribution:** 3
**Rating:** 6
**Confidence:** 3

**Summary:**

The paper studies polysemantic neurons and defines what a representational footprint is in terms of how it can be attributed to the previous layer, as well as clustering these footprints to disentangle different concepts. They find that different architectural inductive biases can lead to different mechanisms by which models form concepts, with a unique U-shaped trend in conceptual diversity in vision transformers compared to CNNs.

**Strengths:**

The authors provide an original and practical mechanism for disentangling polysemantic neurons. They highlight how their approach compares to existing techniques like SAEs or circuits-based interpretability. Their diagrams intuitively demonstrate how a neuron's representational footprints can be separated into different concepts and clusters. They show a clear trend difference between different architectures, and provide an interesting insight into training decisions.

**Weaknesses:**

The focus on images and image models may bring in biases regarding visual separability over other relevant semantics that we might care about when interpreting models broadly. The choice to only compare against PURE and fix the number of clusters means it is difficult to get a sense for how robust the result. While this is a limit of PURE, more ablations could be beneficial. Furthermore, the paper does not expand on its hypothesis that architectural inductive biases matter, and more research could be done to validate this.

**Questions:**

1. How robust are the results across different attribution methods for selecting clusters?
2. Can you compare this to the baselines of using other interpretability techniques?
3. Does this generalise for larger ViTs/ConvNets as well as other pre-training regimes?

---

### Note · Authors · 2025-11-14

I have read and agree with the venue's withdrawal policy on behalf of myself and my co-authors.